# Sub-wavelength patterned pulse lithography for efficient fabrication of large-area metasurfaces

Lingyu Huang[1,3], Kang Xu [1,3], Dandan Yuan[1], Jin Hu[1], Xinwei Wang[2] & Shaolin Xu [1]✉

Rigorously designed sub-micrometer structure arrays are widely used in metasurfaces for light modulation. One of the glaring restrictions is the unavailability of easily accessible fabrication methods to efficiently produce large-area and freely designed structure arrays with nanoscale resolution. We develop a patterned pulse laser lithography (PPLL) approach to create structure arrays with sub-wavelength feature resolution and periods from less than 1 μm to over 15 μm on large-area thin films with substrates under ambient conditions. Separated ultrafast laser pulses with patterned wavefront by quasi-binary phase masks rapidly create periodic ablated/modified structures by high-speed scanning. The gradient intensity boundary and circular polarization of the wavefront weaken diffraction and polarization-dependent asymmetricity effects during light propagation for high uniformity. Structural units of metasurfaces are obtained on metal and inorganic photoresist films, such as antennas, catenaries, and nanogratings. We demonstrate a large-area metasurface ($10 \times 10$ mm$^2$) revealing excellent infrared absorption (3–7 μm), which comprises 250,000 concentric rings and takes only 5 minutes to produce.

Recently, metasurfaces for light modulation have attracted significant attention due to the increasing demand for miniaturized ultra-thin optical systems[1-4] with functions such as beam shaping (deflection, focusing, vortex modulation)[2,5-8], electromagnetic cloak[9-11], polarization control[12], perfect absorption[5,13-15], and precise achromatization[8,16-18]. Various structural units on metasurfaces perform crucial functions such as H shapes for directional surface wave coupling to absorb the light or modulate the wavefront[19-22], cross shapes to adjust the chromatic dispersion or absorption[23-25], and catenary shapes to offer high diffraction efficiency and linear proportion between period and phase delay[26,27]. A combination of several units with continuous phase modulations is a critical type of metasurface for light splitting and deflection, including split rings with various directions[28,29], antennas with multiple angles of spinning or opening[4,28,30,31], squares or circles with various sizes[32-34]. It is also

helpful to achieve broadband modulation by integrating several structures with different scales or shapes[15]. Structural units with tens-of-micrometer scale period usually have long responded wavelength from mid to far-infrared wavelength (such as THz frequency), which have received great attraction on surface-enhanced infrared absorption (SEIRA) spectroscopy, cloaking, and THz communication[10,25,28,32,35].

The development of fabrication methods, especially mass production and easily accessible methods of metasurfaces, falls behind their designs. Electron beam lithography and extreme ultraviolet lithography with outstanding sub-10-nm resolution are general approaches with high freedom in achievable structures to verify the metasurface design for different light modulation abilities. Recently, direct laser writing as a maskless lithography method has been gradually applied in micro-/nanofabrication, whose resolution achieves

[1]Department of Mechanical and Energy Engineering, Southern University of Science and Technology, 1088 Xueyuan Avenue, Shenzhen 518055, China. [2]Department of Mechanical Engineering, Iowa State University, Ames, IA 50011, USA. [3]These authors contributed equally: Lingyu Huang, Kang Xu. ✉e-mail: xusl@sustech.edu.cn

hundreds of nanometers and is mainly restricted by the diameter of the laser spot and the sensitivity of the photoresist. The manufacturing speed and graphic precision of maskless lithography by direct writing depend entirely on the velocity and precision of the motion stage[36]. Referring to practical research and applications of metasurfaces, it is needed to develop a low-cost, highly efficient, and easily accessible method to fabricate large-area metasurfaces with various rigorously designed arrayed structures. Nanoimprinting has been applied in large-area metasurface processing, which hugely relies on the resolution of the mask and may lose the precision of the structures due to defects generated during duplications[37–39].

Besides maskless lithography and mask-dependent duplicating methods, a modulated laser beam with one-step patterning ability has attracted researchers' concentration. The computer-generated hologram (CGH) is a commonly used phase diagram for wavefront modulation of light through an SLM, usually calculated by the iterative Fourier transform algorithm, also called the Gale-Shapley algorithm[40,41]. Whereas CGH makes it challenging to produce microscale structures with sub-wavelength scale features and several micrometers period. The disordered phase distribution in the wavefront modified by CGH would highly disturb the energy distribution during light propagation (with comparison shown in Supplementary Fig. 1)[42,43]. Direct phase masks with binary gratings to remove the background show similar limitations due to strong diffraction at the boundary and uneven intensity inside the wavefront[44,45]. Simple binary masks with polarizers to modulate the wavefront intensity are concise and capable of generating a wavefront with consistent phase distribution, which is still not good enough in ablation performance on the bulk materials because intense ablation would amplify the shortcoming of energy unevenness in the wavefront[46].

In this work, we propose a patterned pulse laser lithography (PPLL) method under ambient conditions to realize the highly efficient production of large-area metasurface composed of periodic units with a period from sub-micrometer to tens of micrometers, highly alterable shapes, and uniform morphology with sub-wavelength feature size. We duplicate the same pattern in the wavefront by separating pulse ablation on substrates which enables the efficient fabrication under a high laser repetition frequency. Metal and inorganic photoresist films with ablated or phase-modified patterns could form metasurfaces or act as masks for fabricating structural units on metasurfaces. The critical step is to modulate the pulse with a uniform patterned appearance with circular polarization following the pixeled phase delays from a spatial light modulator (SLM) with gradient-grayscale boundaries. The phase masks with gradient-grayscale boundaries result in more uniform energy distributions at the wavefront by weakening diffraction during light propagation, which helps to reach smaller feature sizes simultaneously. The circular-polarized wavefront is employed to improve the consistency of patterns' linewidth in different directions, especially for near-diffraction-limit width. Compared with existing methods, PPLL could be applied to various materials such as metals and create the unit pattern with sub-wavelength feature size in one step, ensuring consistency and accuracy.

## Results and discussion
### The principle of the PPLL process
As shown in Fig. 1a, a linear polarized femtosecond laser with a center wavelength of 520 nm and a repetition frequency of 10 kHz is modulated with the polarization distribution and then filtered to remain the homogeneous linear polarization, which would be turned to circular polarization by a QWP, and achieve a tailored patterned light intensity finally (with illustrations of beam profiles in Fig. 2a–e). The phase delay does the modulation through a reflection-type phase-only SLM with patterned quasi-binary phase masks, which comprise two major phase delay values with gradient boundary (Fig. 1c). Patterned pulses are continuously imposed into the samples and spatially separated during

the high-speed scanning, as shown in Fig. 1b to fabricate structural arrays with sub-wavelength feature size and fine recast layers on boundaries, as illustrated in Fig. 1c–e. The film's ablation thickness and modification depth depend on pulse energy and material properties. The relationship between desired scanning speed, period, and laser repetition frequency can be written as follows:

$$D = \frac{v}{f} \tag{1}$$

where $v$ is the scanning speed, $D$ is the pattern period, and $f$ is the laser pulse repetition frequency. Every pulse or group of pulses results in an ablation or phase-modification region following the shape of the tailored intensity.

### Grayscale gradation of phase mask boundary to improve ablation precision
Patterned pulse with gradient intensity boundary achieved by the phase mask with gradient grayscale boundary on SLM can improve the precision and linewidth resolution of the ablated shape.

The gradation is created by polarization filtering due to gradient phase delay from phase mask with processing schematic in Fig. 2a–e that corresponds to the positions 1–6 in Fig. 1a (with a detailed process in Supplementary Fig. 2). The pattern information is transferred from the phase mask on SLM to the polarization distribution of the wavefront, which is then transferred into intensity amplitude by the polarization filter. It is noticed that the elliptical polarization delivers the information of gradient boundary. Further theoretical calculation and analysis are stated in the supplementary information's second section. A 50× objective (NA = 0.55) is applied to produce triangles (Fig. 2) and L shapes (Fig. 3) to study the related effects on a larger scale with a more obvious phenomenon. Heavy diffraction happens when geometric patterns with sharp angles are imposed (Fig. 2f4). Thus, pure binary phase masks such as triangles result in a rough ablated pattern boundary (Fig. 2f2). The strong diffraction comes from an abrupt change in light intensity at the boundary due to Fresnel approximation. Specifically, the Fresnel approximation derived from the first Rayleigh-Sommerfeld diffraction with a paraxial situation that is given by[47]:

$$\begin{cases} U(x_2,y_2,z) = \frac{i\exp(-ikz)}{\lambda z} \iint_{-\infty}^{+\infty} U_0(x_1,y_1,z_1)\exp\left\{\frac{-ik}{2z}\left[(x_2-x_1)^2+(y_2-y_1)^2\right]\right\}dx_1dy_1 \\ U_0(x_1,y_1,z_1) = I_0\exp\left(\frac{-(x_1^2+y_1^2)}{\omega^2(z_1)}\right) \\ I_0 = \begin{cases} 1(\sigma=0) \\ 0(\sigma=\pi) \end{cases} \end{cases} \tag{2}$$

where $z$ is the distance from the diffraction screen (just after the polarizer), $\lambda$ is the wavelength of the light, $k$ is the wave vector of the light, $\omega$ is the waist radius, $\sigma$ is the phase on the mask as defined in Fig. 1c, and $U$ is the light intensity distribution. Significantly, the $U_0(x_1, y_1, 0)$ denotes the original input light, namely the superposition of a Gaussian beam and the phase masks in our cases. This Fresnel approximation fits great when the Fresnel number $N \geq 1$, which is defined as:

$$N = \frac{\pi a^2}{\lambda z} \geq 1 \tag{3}$$

where $a$ is the width of the whole structure in the phase mask. According to our experimental conditions (with $\lambda$ of 520 nm and $z$ of around 1000 mm because that QWP and polarizer are placed as closely as possible to the lens1), the $a$ is supposed to be larger than 721 μm (around 58 pixels) that is always satisfied in our experiments so that all calculations on light propagation are processed by the Fresnel approximation.

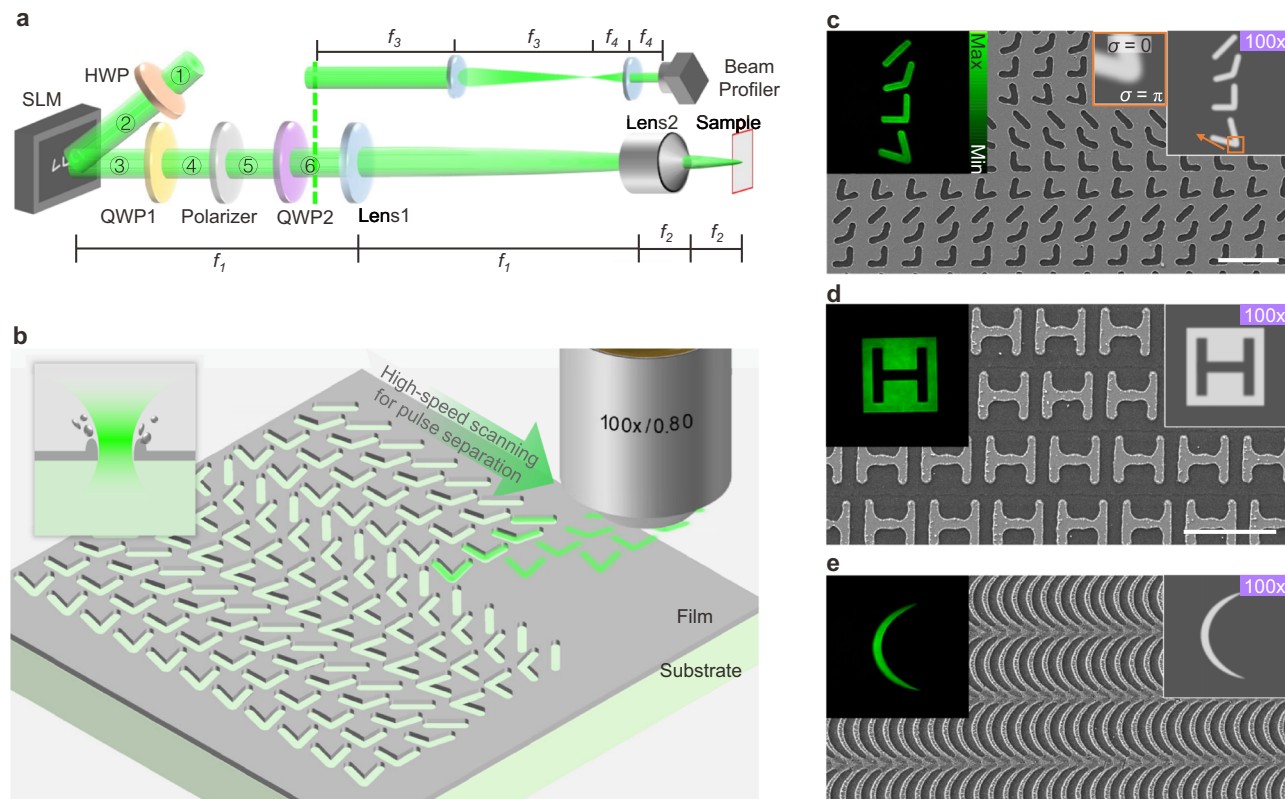

**Fig. 1 | Schematics of the patterned pulse laser lithography (PPLL) process and the scanning electron microscopy (SEM) images of the results. a** The PPLL system (HWP: half waveplate, QWP: quarter waveplate, $f_1 = 1000$ mm, $f_2 = 2$ mm for lens2 of 100× objective or 4 mm for lens2 of 50× objective, $f_3 = 500$ mm, $f_4 = 10$ mm). **b** Schematic of high-speed scanning and separated pulse ablation for PPLL. **c, d, e** The quasi-binary phase masks on SLM (right up corner) with corresponding energy distributions of the modulated pulse (left up corner, detected by a beam profile) and the ablation demonstrations containing antennas by pulse energy of 0.15 μJ, H shape units by pulse energy of 0.17 μJ, and catenaries by pulse energy of 0.57 μJ on samples. The 100× marks refer to the 100× objective as lens2. For **c, d** the substrates comprise 10 nm Au, 3 nm Cr and a fused silica wafer from top to bottom. For **e** the substrates comprise 30 nm Cr on a fused silica wafer. Scale bars: 5 μm.

Here, 6 pixels of gradient grayscale (with approximate sine distribution) at every boundary of the phase masks are suitable, which could form efficient grayscale gradation but not affect the structural precision of the shape. Comparing experimental results and propagation calculation results in Fig. 2f1, f4, 2g1,g4, the addition of grayscale gradation could smooth the change of phase and intensity to weaken the diffraction, which helps achieve the patterns with straight boundaries and sharp angles. The curvature radius of the sharp angle could decrease from 367 nm to 119 nm, namely a sharper shape, as shown in Figs. 2f2, g2. It is noticed that the slight asymmetry of two 45° angles comes from comprehensive errors due to the spherical aberration of the SLM surface, imperfect Gaussian distribution of incident light, and the error in center misalignment, which cannot be completely avoided.

In addition to regulating the pattern appearance, gradient grayscales along the mask boundary can reduce the linewidth even beyond the diffraction limit. As shown in Supplementary Fig. 3, a sharp L shape with a width of 12 pixels cannot be ablated completely. Under the same laser pulse energy, a sharp L shape with a width of 20 pixels could be ablated totally. By comparison, the 20-pixel line with 6-pixel gradation on each boundary is feasible to achieve smaller line width than a 20-pixel sharp one.

### Uniform ablation by considering polarization-dependent effects and isotropy of structures

A polarization-dependent asymmetrical ablation effect impacts metal film ablation morphology caused by polarization-dependent diffraction[48]. We replace QWP2 in Fig. 1a with an HWP to tune the light polarization to 0°, 45°, and 90°. Stronger ablation (in the area with high light intensity) happens along the rod perpendicular to the polarization leading to a larger linewidth, which is evident in the SEM images (Fig. 3a,c). By comparison, the results ablated by laser with 45° and circular polarization show equal width along both rods of two directions. We consider a vector model of the first Rayleigh-Sommerfeld diffraction with a paraxial situation to consider the polarized laser propagation as stated in Section 4 of supplementary information. According to Fig. 3f, the electric field intensity in the z-direction (as the light propagation direction) reveals an intensity concentration in the polarization-perpendicular rod resulting in stronger ablation and larger linewidth. This slight difference becomes more evident when the linewidth is relatively small, especially when the linewidth is close to the diffraction limit. Thus, the circular polarization (Fig. 2e) would be more suitable for random patterns due to the same polarization components in all directions. In contrast, linear polarization could be applied for unique patterns such as 1D gratings.

In addition, isotropic shapes such as ring shapes reveal excellent shape stability during light propagation, keeping ablated structures uniform. As shown in Fig. 3g–j, the ultra-uniform wavefront with harmless concentric diffraction for structural regulation, could result in isotropic structures with regular shapes and relatively smaller feature sizes of 303 nm, approximately 76% of the diffraction limit (397 nm under NA of 0.8).

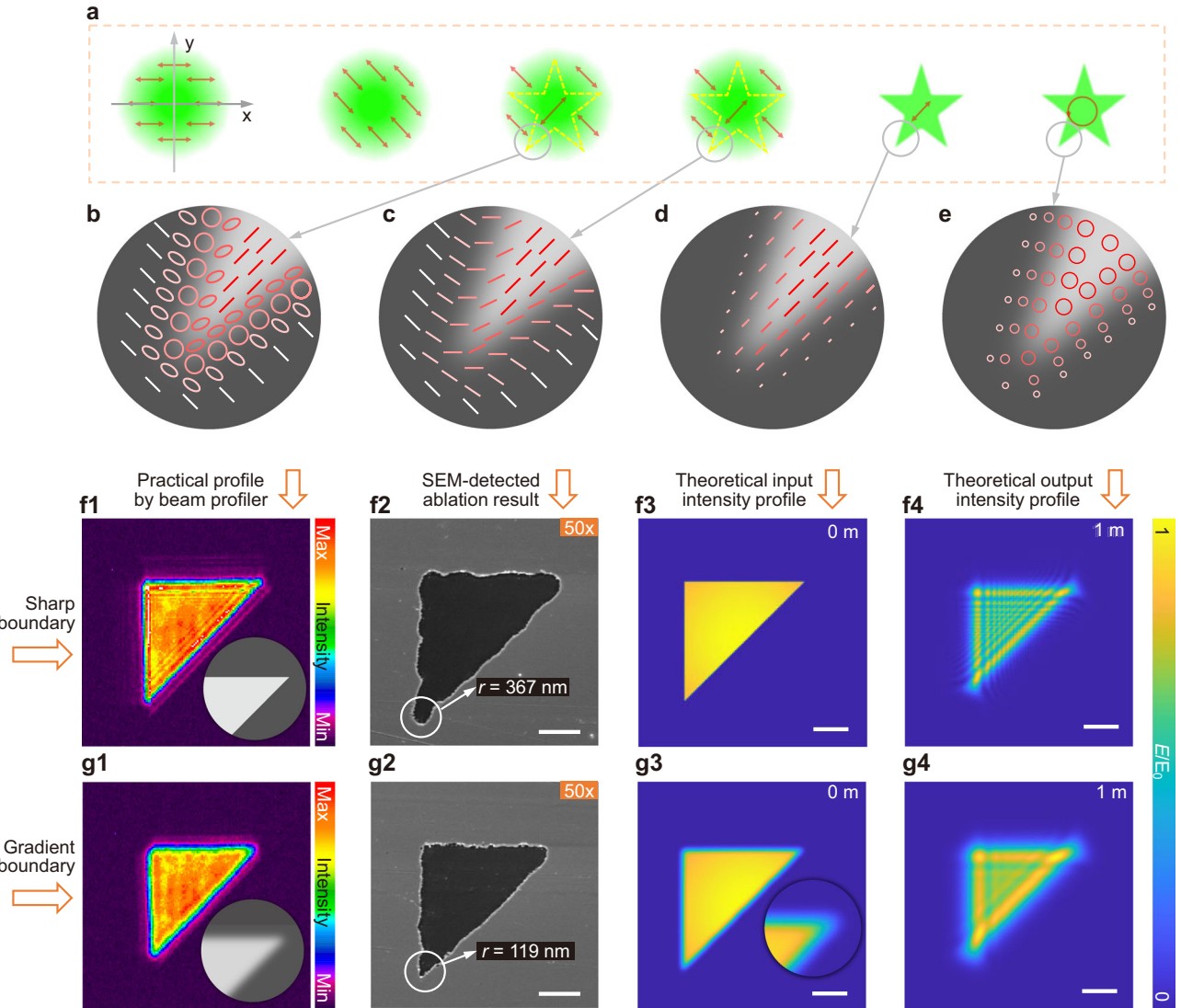

**Fig. 2 | The improvement of fabrication morphology by gradient grayscales on the pattern boundaries in the phase masks. a–e** Schematic of the light modulation process for PPLL resulting in patterned pulse with gradient boundary energy distribution. The arrows indicate the polarization direction. **f, g** The detected light intensity profile with the phase mask inserted, SEM images of ablation results with pulse energy of 0.24 μJ, and theoretically input intensity and output intensity at the ablation position in light propagation calculation for sharp-boundary triangle mask and gradient-boundary triangle mask, respectively. The 50× marks refer to a 50× objective as lens2. Scale bars: 2 μm.

## Linewidth-pixel relationship and sub-diffraction-limit resolution of PPLL

We could predict the linewidth of structures fabricated by PPLL through magnification analysis. As shown in Fig. 4a, the ablated linewidth is nearly proportional to the pixel numbers in phase masks. We fabricate all the single lines with pulse energy just above the ablation threshold. Referring to Supplementary Fig. 4 and the corresponding analysis, the ablated linewidth $L$ follows the following equations:

$$L = \frac{N \times l}{f_1/f_2} \times k \tag{4}$$

where $N$ is the number of pixels, $l$ is the pixel size of liquid crystal pixel in SLM (equal to 12.5 μm for the device we use), and $k$ (from 0 to 1) is the ratio of beam width to ablated region width which is related to the ablation/modification threshold of the sample, laser fluence, and pulse intensity distribution. We obtain the $k$ with around 0.96, and the $k$ under higher NA reveals higher amplitude due to a more intense laser focus.

The relationship between linewidth and pixel number does not follow the linear proportion when the pixel number is smaller than 35. The linewidth shows few changes when the pixel number is smaller than 20, owing to the linewidth being close to the diffraction limit (397 nm for NA = 0.8 of 100× objective, the purple dashed line in Fig. 4b). And the parameter $k$ gradually reduces to around 0.8 due to the narrowing down of the linewidth with a sharper intensity peak resulting in linewidth beyond the diffraction limit. Here we demonstrate a sub-wavelength grating with a period of ~450 nm by a line-shape laser pulse (Fig. 4c–e).

## Demonstrations and applications of patterned structures

The PPLL method could produce rigorously designed patterns on film samples (with demonstrations in Supplementary Figs. 4, 5, 7, 8). Furthermore, the nanopatterned films on substrates processed by PPLL can act as masks for further etching to fabricate structures with a larger depth-to-width ratio on metasurfaces for better light modulation efficiency.

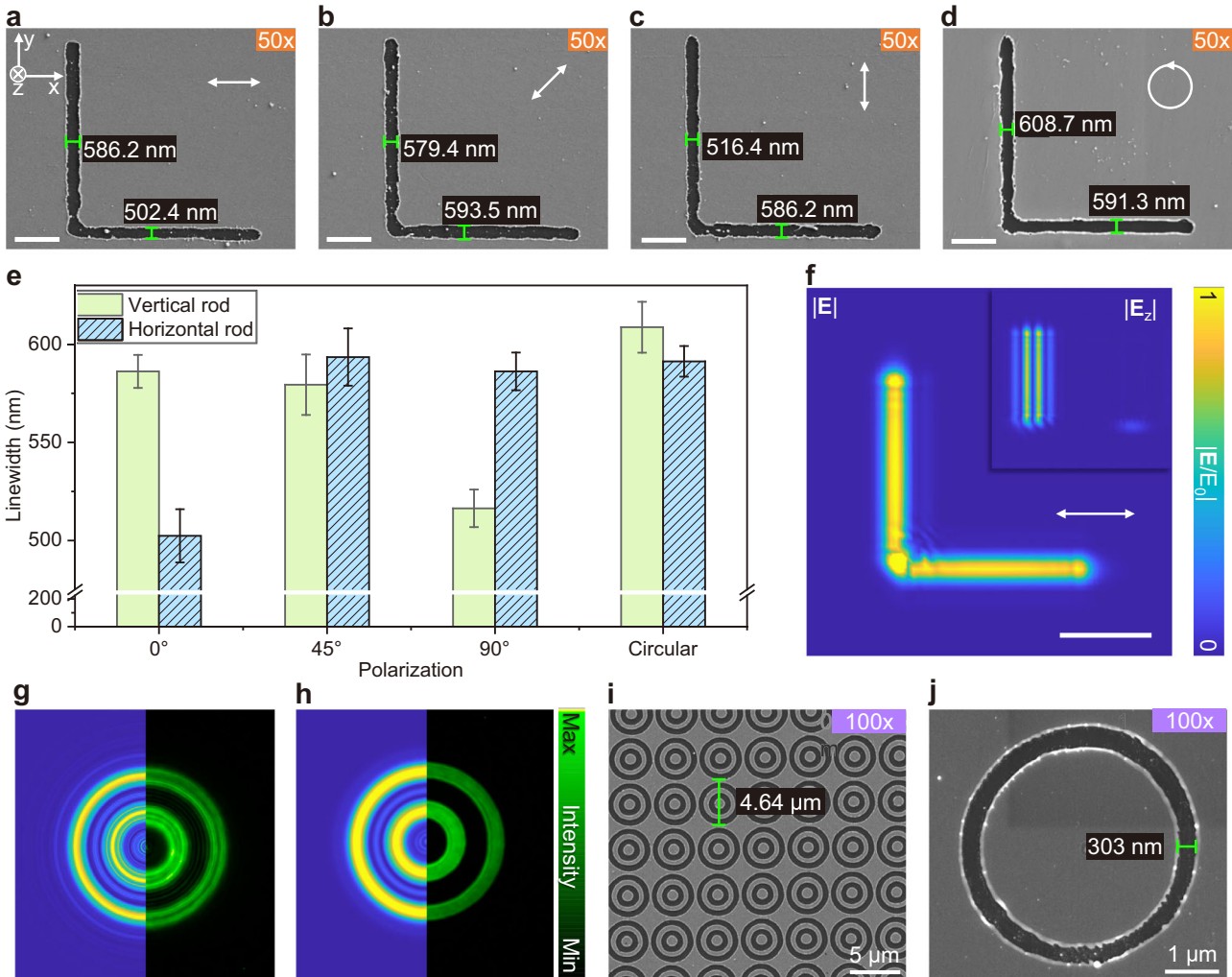

**Fig. 3 | The L shapes produced by PPLL with different polarizations and isotropic rings made with circular polarization. a–d** Ablation result of the gradient boundary L shape with 20-pixel width (with 6-pixel gradation on each boundary) by 0°, 45°, 90°, and circular polarization, respectively. The white arrows indicate the polarization direction. The applied pulse energy is 0.15 μJ. Scale bars: 2 μm. **e** The measurement of horizontal and vertical rods' linewidths with different polarization states. The error bars represent the standard deviation of measured linewidth. **f** The simulated propagation results of the L structure with 0° polarization. The polarization direction indicated as the arrow shows. Scale bar: 2 μm. **g, h** The simulated propagation results and detected beam profiles of concentric rings without and with gradient boundary. **i** concentric ring arrays by pulse energy of 0.23 μJ. **j** Single rings with a width of 303 nm by pulse energy of 0.09 μJ. The samples are composed of 10 nm Au, 3 nm Cr on SiO₂ substrate.

We demonstrate several fundamental nanostructure arrays for metasurfaces by PPLL with additional plasma and chemical etching, as shown in Fig. 5. In Fig. 5b–g and Supplementary Figs. 5 and 6, we illustrated structure arrays after inductively coupled plasma (ICP) etching with the laser patterned Cr films as masks on SiO₂ samples (Fig. 5a). Figure 5b shows an H shaped structure array after reverse H shaped pulse laser ablation and plasma etching, which is generally applied to manipulate phase response under microwave even THz[19,22]. Figure 5d shows classical V-shaped antennas for regulating electromagnetic waves by resonance[49]. Each group of four V-shaped antennas with different orientations is obtained by single patterned pulse ablation. Figure 5e reveals regular concentric rings with high uniformity. Catenaries and gratings are also demonstrated in Fig. 5f, g, which are common structures for metasurface offering high diffraction efficiency[26,27,50–54]. Supplementary Fig. 5 exhibits catenaries and gratings with smaller linewidth, periods, and larger height-to-width ratio and 2D gratings. When the reverse patterned pulse laser is imposed to create the male structural arrays (Fig. 5b–d and Supplementary Figs. 6a, b, d), a slight overlap between pulses is required. Remarkably, the sub-wavelength

resolution has been achieved in the linewidth of all the structural units.

Apart from the direct laser ablation, phase change material Ge₂Sb₂Te₅ (GST) is available to fabricate stereoscopic structures by chemical etching after patterned laser modification, which could offer an excellent performance of the wavefront modulation due to structural depth up to hundreds of nanometers and a high refractive index of GST[55]. The GST material has been widely applied for tunable metasurfaces owing to its phase-dependent optical property, robust reproducibility, and quick crystal-phase changing[56,57]. It is also a frequently used inorganic phase-change photoresist for micro-/nanofabrication due to the different etching rates between amorphous (GSTₐ) and crystallized (GSTᴄ) phases (Fig. 5h)[58]. As illustrated in Fig. 5i–l, we demonstrate four types of common three-dimensional structures on GST films (the original samples with 250 nm GST film deposited on SiO₂ substrates). As shown in Fig. 5i, we fabricate an array of a combined unit composed of ring and cross structures with different dimensions resulting in a combination of different resonance frequencies, which is helpful for broadband absorbers[15]. Furthermore, we demonstrate chiral structures, which are always utilized to adjust

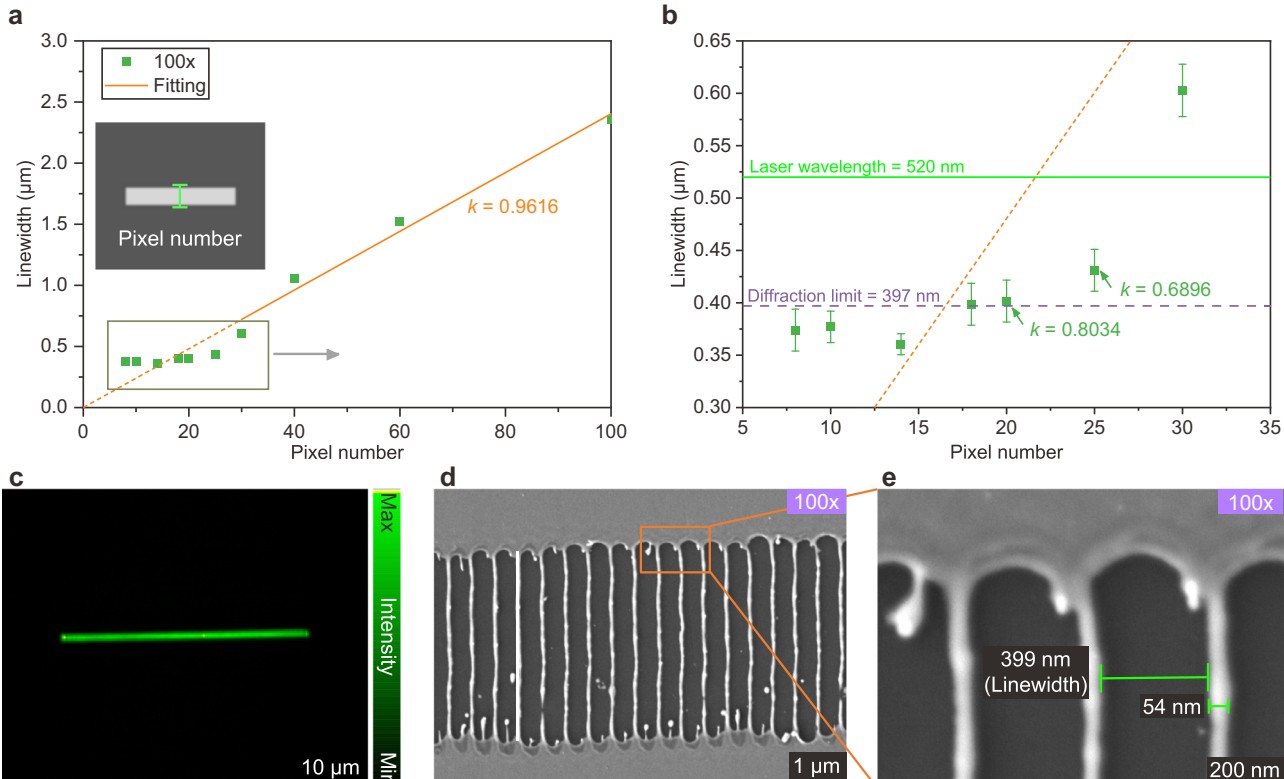

**Fig. 4 | Line-shaped pulse laser ablation with different pixel numbers of linewidth. a, b** Statistics of ablated linewidth under various pixel numbers with diffraction-limit at 397 nm (purple dashed line) and wavelength at 520 nm (green solid line). The error bars represent the standard deviation of measured linewidth. **c** Beam profile of the line pulse modulated by a phase mask with 14-pixel linewidth. **d, e** Gratings produced by PPLL with a period of 453 nm and remaining linewidth of 54 nm (laser-ablated linewidth is 399 nm) by pulse energy of 0.07 μJ.

the wavefront because of their selective response to the left or right circular polarization (Fig. 5j)[59]. Besides, concentric structures with open ends are produced with excellent uniformity and precision, which are great resonant antennas for absorbers (Fig. 5k, l)[28,60].

Above all, the PPLL can produce various types of metasurface units directly or by combining with etching processes with high efficiency that reveals excellent potential in the fabrication of practical large-area metasurfaces. The structure size of one unit can be precisely controlled from less than 1 μm up to tens of micrometers by the zooming out mechanism. The depth-to-width ratio can even reach around 7:1 under a linewidth of 92 nm (Supplementary Fig. 5f).

It is noticed that the high resolution and positioning accuracy of the displacement stage are significant for large-area repeatability. Under a stable device condition, we can slightly adjust scanning speed to modulate the period distribution and alignment (Supplementary Figs. 7 and 8), revealing further controllability in the spatial distribution of structural units. The gratings in Fig. 5g come from the seamless join between line pulse scanning with a proper scanning speed.

According to the results in Fig. 3, the PPLL is quite suitable for fabricating uniform concentric rings, which could be structural units for electromagnetic resonance forming metasurface. We design two concentric rings by the Finite-difference time-domain (FDTD) method (inserted in Fig. 6d) to achieve absorbers with different absorption wavelengths on the mid-infrared wavelength of 3−7 μm (3333−1429 cm⁻¹), which can be further applied in SEIRA for biology detection. As shown in Fig. 6a, a classical sandwich structure was designed to improve the absorption efficiency on the resonant wavelength. We use the evaporated $SiO_2$ layer with a thickness of 50 nm as the insulator layer, the Au layer with a thickness of 70 nm for the reflective metal layer at the bottom, and patterned GST with a total thickness of 250 nm for the resonant structures. The resulting

structure depth reaches ~200 nm after PPLL and chemical etching, as shown in Supplementary Fig. 9, providing efficient resonance conditions for metasurface.

According to the sectional electric field distribution in Fig. 6d, the light intensity accumulates in the $SiO_2$ insulator layer under a TM-polarization mode excitation. And the double-valley reflectance of small concentric rings mainly comes from resonance in the center circle and two outer rings, respectively. The large concentric rings only exhibit a single resonance in the center circle. After processing with the PPLL method and further chemical etching, we produced two concentric rings with different dimensions (Fig. 6c), whose absorption peaks match our design with less than 300 nm shift of spectrum valleys with excellent peak absorbance over 98%.

In summary, we developed a patterned pulse laser lithography (PPLL) to produce metasurfaces composed of sub-wavelength structural units with high efficiency and low cost at ambient conditions. The principles of PPLL have been studied, and the fabrication processes have been optimized. Boundary gradation of quasi-binary phase masks and circular polarization can help weaken wavefront deformation from diffraction and polarization-dependent asymmetricity effect during light propagation, resulting in uniform structures and sub-wavelength feature size. We obtained various patterns containing classical sub-wavelength structures for metasurface with the minimum feature size beyond the diffraction limit. Combining with etching processes, we demonstrate an efficient large-area metasurface as a mid-infrared absorber that achieves peak absorption over 98% under the wavelength of 3−7 μm. This novel fabrication technique is highly feasible to obtain rigorously designed patterns with nanoscale feature size and high efficiency on various materials. The consistency and accuracy of the designed patterns are highly guaranteed by the one-step separated modulated pulse energy deposition by direct high-speed laser

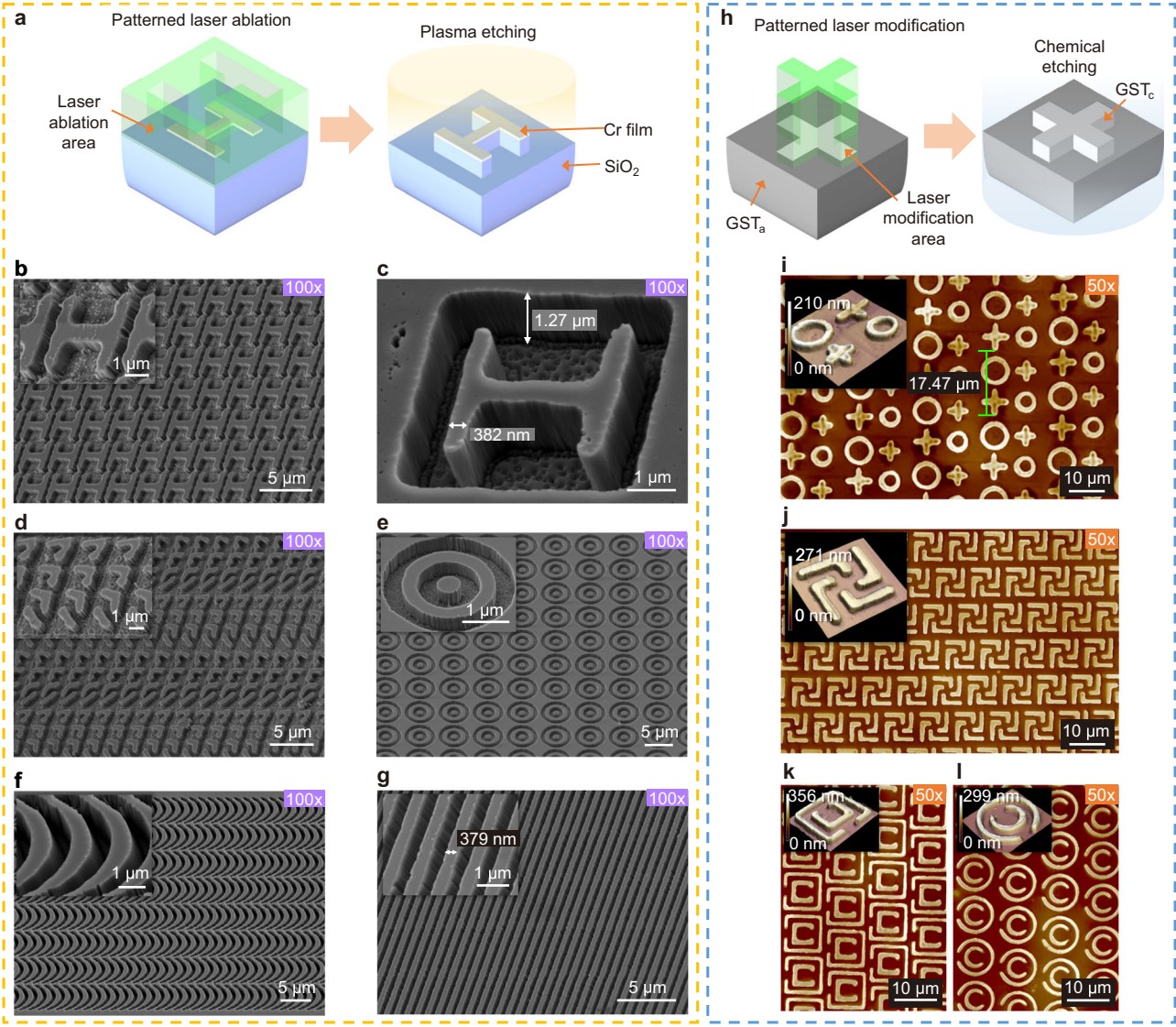

**Fig. 5 | Classical patterns for metasurface achieved by the PPLL. a, h** Schematics of fabrication processes of structures by etching after PPLL. **b–g** SEM images of the patterns after etching, including H shapes, several antennas with different orientations in a unit, concentric rings, catenaries, and gratings (with patterned 30 nm Cr as masks on fused silica substrate). **i–l** 3D demonstrations of the patterns including geometric combination, chiral shape, open-ended concentric circular/rectangle ring on Ge$_2$Sb$_2$Te$_5$ (GST) film with SiO$_2$ substrate after chemical etching. 12-pixel gradation is applied in all phase masks.

scanning. Patterns with higher resolution would be achieved by further adoption of shorter laser wavelength (smaller diffraction limit), SLM with smaller pixel size ($l$), 4 f system for beam narrowing (larger $f_1/f_2$), and photoresist samples with smaller ablation/modification threshold. The PPLL method promises broad applications for large-area metasurface to achieve perfect absorption, polarization filtering, and beam deflection.

## Methods
### Sample fabrication

(1) The high-speed scanning is processed by ultra-precision linear motor stages (XMS100-S, Newport) with a maximum moving speed of 300 mm/s. The phase delay does the light modulation through a reflection-type phase-only SLM with phase masks (LCOS-SLM X15213-16, Hamamatsu, pixel number of 1272 × 1024, pixel width of 12.5 μm for each pixel, 8 bit of 256 grayscale levels, fill factor 96%). A linear polarized femtosecond laser (Spectrum Physics, with a center wavelength of 520 nm, a pulse width of

300 fs, a repetition frequency of 10 kHz, and maximum pulse energy of 19 μJ) is applied to ablate samples.

(2) All original substrates are first processed by ultrasonic cleaning with alcohol and isopropanol for 10 minutes, respectively. We deposited 3 nm Cr and 10 nm Au successively on cleaned fused silica (JGS1) substrates (Figs. 1c, d, 2–4) and 30 nm Cr on fused silica substrates (Figs. 1e and 5b–g) by electron-beam evaporation (EBE, TF500, HHV). Besides, we deposited 250 nm GST on cleaned quartz substrates by magnetron sputtering (KYKY-500CK-500ZF) as samples in Fig. 5i–l. The samples in Fig. 6 contain four layers. Firstly, 5 nm Cr and 70 nm Au were deposited on Si (crystalline orientation (001), N-type doping) substrates by EBE, then 50 nm SiO$_2$ and 250 nm GST were deposited on top of the Au layer by magnetron sputtering.

(3) The plasma etching process in Fig. 5b–g is executed by ICP etching equipment (GSE200Plus, NMC). The etching gas consists of 40 sccm CHF$_3$, 10 sccm SF$_6$, and 10 sccm O$_2$ in 8 mTorr chamber pressure, etching at 400 W SrcRF power and 50 W BiasRF power for 400 s (resulting in a thickness of 0.7 μm in Fig. 5b, d, e) or 720 s

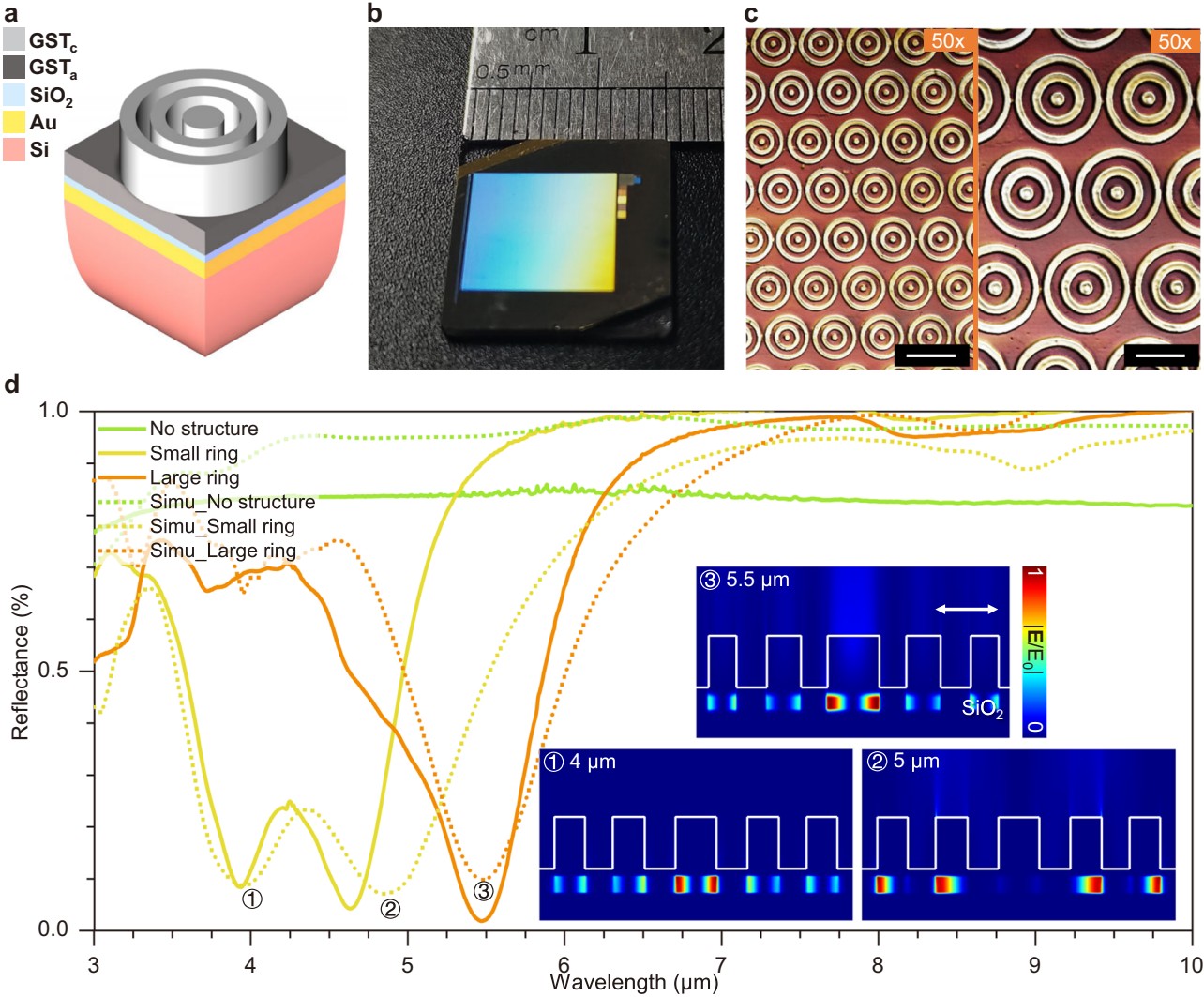

**Fig. 6 | The application of concentric rings in metasurface as a mid-infrared absorber. a** Schematic of the designed structure for the absorber. **b** Photograph of the large-area metasurface with concentric rings as structural units. **c** Surface morphology of small (left) and large (right) concentric rings produced by 50× objective. Scale bars: 10 µm. **d** The reflectance spectrum for the experimental (solid line) and simulation (dashed line) data for the absorber with a small ring structure array (yellow), large ring structure array (orange), and without structures (green) on the top GST layer. The inserted sectional view of |*E*/E| in the sectional view reveals the resonant mode under three typical wavelengths. The white arrow indicates the polarization direction.

(resulting in a thickness of 1.3 µm in Fig. 5c, f, g). The etching time does not attain the etching limitation of 30 nm Cr.

(4) The chemical etching is processed by using an alkaline etchant (25% Tetramethyl ammonium hydroxide (TMAH) solution) in Figs. 5i–l and 6.

### Absorbance spectrum
The reflection spectrum of the sample was detected by Fourier transform infrared spectrometer (VERTEX 70, BRUKER) incorporated with a Bruker IR microscope (HYPERION 2000, BRUKER) among the wavelength of 3–10 µm.

### Sample characterization
The SEM images were taken by field emission scanning electron microscopy (SEM, Merlin, ZEISS). The pictures of the laser profiles were taken by the beam profiling camera (BeamGage LT665, Ophir-Spiricon). And the pulse energy was measured by a power meter (1919-R, Newport) with a matching probe (919P-003-1, Newport) after the polarizer (fifth position in Fig. 1a). The surface morphologies in Fig. 5i–l and Fig. 6c were detected by laser scanning confocal microscopy (LSCM, VK-X1000, KEYENCE). The height of the stereoscopic structure

in supporting information was characterized by an atomic force microscope (AFM, Dimension EDGE, Bruker).

### Diffraction calculations and FDTD simulations
The calculation for diffraction theory was processed by MATLAB (R2019b). The optical simulation for the absorber design was processed by the FDTD commercial software (Lumerical 2020 R2, ANSYS).

## Data availability
All data are available within the Article and Supplementary Files, or available from the corresponding authors on request.

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

## Acknowledgements

This work was financially supported by Guangdong Provincial University Science and Technology Program (Grant No. 2020KTSCX119) and Shenzhen Science and Technology Programs (Grant No. 202009251-55508001, GJHZ20190820151801786, KQTD20170810110250357). Besides, SEM data were obtained using equipment maintained by the Southern University of Science and Technology Core Research Facilities.

## Author contributions

L.H. and K.X. contributed equally as the first authors to experiment, characterization, and paper writing. D.Y. performed the chemical etching of GST samples and improved the analysis of the absorber. J.H. carried out the plasma etching process. X.W. reviewed and improved the manuscript. S.X. conceived and supervised the project. All authors contributed to the idea discussion and paper writing.

## Competing interests

The authors declare no competing interests.
