## [Peer Review File · Nature Communications]

Sub-wavelength patterned pulse laser lithography for efficient fabrication of large-area metasurfacesREVIEWER COMMENTS

Reviewer #1 (Remarks to the Author):

The manuscript considered herein introduces the technique of patterned pulse laser lithography (PPLL) as an alternative nanofabrication tool for the production of arrays with sub-wavelength feature resolution and periods of several μm on large-area thin films. The separated ultrafast laser pulses feature patterned wavefronts through the use of SLMs under high speeds. The patterned pulses are used to ablate metallic masks of inorganic resists that then can be transferred to a substrate by means of conventional dry etching processes. The manuscript is clear and nicely organized from the description of the technique until the provision of a series of examples of the structures that can be produced. The results are sound and I am happy to recommend this work for publication after a few points can be clarified by the authors.

- The authors provide some context for their approach in the introduction of the work. However, I believe the authors should include a more detailed comparison between the PPLL and the traditional laser writing technique available in most clean rooms, since both cases require a moving stage and need subsequent RIE processes to transfer the pattern to the substrate. What would be the advantages of PPLL? I believe this comparison would highlight the advantages of their proposed technique. Also, the comparison with ebl and nanoimprinting techniques should also include a discussion regarding the high resolution that can be achieved with those.

- Figure 4 show a 400nm feature resolution, could this be improved with other laser choices? Would it be possible to achieve samples operating in the visible or NIR region?

- What is the writing speed? is it limited by the SLM?

-

- The authors claim in their manuscript "The nanopatterned films on substrates processed by PPLL can act as masks for further etching to fabricate structures with a larger depth-to-width ratio .." Could the authors elaborate on this statement? Is the higher aspect-ratio achieved thanks to the use of metal masks rather than a feature of PPLL? since the etching step is performed as a later step using RIE.

Reviewer #2 (Remarks to the Author):

Reviewer report on the paper

Sub-wavelength patterned pulse lithography for extremely efficient fabrication of large-area metasurfaces

The authors report on a fabrication method which can be used to pattern surfaces fast and with very good resolution. Although it was very interesting for me to read the paper and to see the results, I doubt that nature communications is the place where these results should be published. The main improvements of the paper are on detailed optical schemes which should be published in an optical journal or alternatively in an journal of applied physics.

Still I would recommend to improve the paper with regard of readability for a general reader:

- in the abstract and in the text the name separated pulses is used. Does it mean single shots?

- I can not find an intensity in the main text

- I do not find a description of Fig 2 a -e

- fig 2 f and g can not be the intensity as written in the text, I guess f2 and g2 are electron microscopy pictures

Responses to Reviewers' Comments

We highly appreciate the reviewers' recognition of our work with valuable comments that are helpful for revising and improving our manuscript. We have made corresponding revisions in the manuscript and supplementary information, which are highlighted in blue. The detailed changes and replies are listed in the following.

Reply to reviewer #1

Reviewer #1 (Remarks to the Author):

The manuscript considered herein introduces the technique of patterned pulse laser lithography (PPLL) as an alternative nanofabrication tool for the production of arrays with sub-wavelength feature resolution and periods of several μm on large-area thin films. The separated ultrafast laser pulses feature patterned wavefronts through the use of SLMs under high speeds. The patterned pulses are used to ablate metallic masks of inorganic resists that then can be transferred to a substrate by means of conventional dry etching processes. The manuscript is clear and nicely organized from the description of the technique until the provision of a series of examples of the structures that can be produced. The results are sound and I am happy to recommend this work for publication after a few points can be clarified by the authors.

Reply:

We highly appreciate the reviewer's high evaluation of our work and recommendation for publication.

- The authors provide some context for their approach in the introduction of the work. However, I believe the authors should include a more detailed comparison between the PPLL and the traditional laser writing technique available in most clean rooms, since both cases require a moving stage and need subsequent RIE processes to transfer the pattern to the substrate. What would be the advantages of PPLL? I believe this

comparison would highlight the advantages of their proposed technique. Also, the comparison with ebl and nanoimprinting techniques should also include a discussion regarding the high resolution that can be achieved with those.

Reply to question #1:

Thanks for pointing out that a more detailed comparison between the PPLL and conventional laser direct writing would help to highlight the advantages of the PPLL. We have added a discussion in the second paragraph of the introduction about laser direct writing, and the resolution of EBL and nanoimprinting. The most remarkable advantage of PPLL is the capability of efficiently fabricating patterns array with sub-wavelength feature resolution and arbitrary shapes. The one-step exposure process makes the PPLL much more efficient than that of laser direct writing point by point. This statement has also been emphasized at the end of the introduction.

- Figure 4 show a 400nm feature resolution, could this be improved with other laser choices? Would it be possible to achieve samples operating in the visible or NIR region?

Reply to question #2:

The PPLL methods do not limit the wavelength of the laser, as long as the SLM or other light modulation devices like DMD are applicable in the corresponding wavelength. We have added further discussion about resolution improvement in the last part of the conclusion.

- What is the writing speed? is it limited by the SLM?

Reply to question #3:

If the structures are consistent in the whole region, the writing speed completely depends on the speed limitation of motion stages. When we fabricate the metasurfaces with varying structures in different regions, the switching of phase diagrams is required for the SLM, whose refresh rate is 60 Hz and would not largely affect the writing speed.

- The authors claim in their manuscript “The nanopatterned films on substrates processed by PPLL can act as masks for further etching to fabricate structures with a

larger depth-to-width ratio ..” Could the authors elaborate on this statement? Is the higher aspect-ratio achieved thanks to the use of metal masks rather than a feature of PPLL? since the etching step is performed as a later step using RIE.

Reply to question #4:

The high depth-to-width ratio mainly attributes to the application of highly etching-resistant Cr films, which benefit from the PPLL methods with wide applicable materials as we can see in Supplementary Fig. 5.

Reply to reviewer #2

Reviewer #2 (Remarks to the Author):

Reviewer report on the paper

Sub-wavelength patterned pulse lithography for extremely efficient fabrication of large-area metasurfaces

The authors report on a fabrication method which can be used to pattern surfaces fast and with very good resolution. Although it was very interesting for me to read the paper and to see the results, I doubt that nature communications is the place where these results should be published. The main improvements of the paper are on detailed optical schemes which should be published in an optical journal or alternatively in an journal of applied physics.

Reply:

We thank the reviewer for the positive evaluation of the contents of our work.

Still I would recommend to improve the paper with regard of readability for a general reader:

- in the abstract and in the text the name separated pulses is used. Does it mean single shots?

Reply to question #1:

The “separated pulses” do mean single shots. We state the “separated pulses” to strengthen the high efficiency from the high-speed scanning.

- I can not find an intensity in the main text

Reply to question #2:

We think that the reviewer pointed out the lacking of pulse energy. We have added the corresponding pulse energy in the captions of figures.

- I do not find a description of Fig 2 a -e

Reply to question #3:

We mentioned “Figs. 2a-e” on page 5 with a detailed description in Supplementary Fig. 2.

- fig 2 f and g can not be the intensity as written in the text, I guess f2 and g2 are electron microscopy pictures

Reply to question #4:

We are sorry that Fig. 2 is misleading. We have added detailed descriptions in Fig. 2 and its caption. The images from left to right refer to the practical profile detected by a beam profiler, ablation results detected by SEM, theoretical input intensity profile, and theoretical output intensity profile.

REVIEWERS' COMMENTS

Reviewer #1 (Remarks to the Author):

The authors have addressed all the comments nicely, including more context to highlight their approach among competing nanofabrication processes. The technique is sound and the manuscript is well written and organized. I recommend this work for publication.